# Sparse Activations for Interpretable Disease Grading

**Kerol R. Donteu Djoumessi**[1,2]                 KEROL.DJOUMESSI@UNI-TUEBINGEN.DE

**Indu Ilanchezian**[1,2]                           INDU.ILANCHEZIAN@UNI-TUEBINGEN.DE

**Laura Kühlewein**[3]                        LAURA.KUEHLEWEIN@MED.UNI-TUEBINGEN.DE

**Hanna Faber**[3]                                 HANNA.FABER@MED.UNI-TUEBINGEN.DE

**Christian F. Baumgartner**[4]                CHRISTIAN.BAUMGARTNER@UNI-TUEBINGEN.DE

**Bubacarr Bah**[5,6]                                         BUBACARR@AIMS.AC.ZA

**Philipp Berens**[1,2,7]                               PHILIPP.BERENS@UNI-TUEBINGEN.DE

**Lisa M. Koch**[1,2]                                     LISA.KOCH@UNI-TUEBINGEN.DE

[1] *Hertie Institute for Artificial Intelligence in Brain Health, University of Tübingen, Germany*

[2] *Institute of Ophthalmic Research, University of Tübingen, Germany*

[3] *University Eye Clinic, University of Tübingen, Germany*

[4] *Department of Computer Science, University of Tübingen, Germany*

[5] *African Institute for Mathematical Sciences (AIMS) South Africa and Stellenbosh University*

[6] *Medical Research Council Unit The Gambia at London School of Hygiene and Tropical Medicine*

[7] *Tübingen AI Center, University of Tübingen, Germany*

**Editors:** Accepted for publication at MIDL 2023

## Abstract

Interpreting deep learning models typically relies on post-hoc saliency map techniques. However, these techniques often fail to serve as actionable feedback to clinicians, and they do not directly explain the decision mechanism. Here, we propose an inherently interpretable model that combines the feature extraction capabilities of deep neural networks with advantages of sparse linear models in interpretability. Our approach relies on straightforward but effective changes to a deep bag-of-local-features model (BagNet). These modifications lead to fine-grained and sparse class evidence maps which, by design, correctly reflect the model's decision mechanism. Our model is particularly suited for tasks which rely on characterising regions of interests that are very small and distributed over the image. In this paper, we focus on the detection of Diabetic Retinopathy, which is characterised by the progressive presence of small retinal lesions on fundus images. We observed good classification accuracy despite our added sparseness constraint. In addition, our model precisely highlighted retinal lesions relevant for the disease grading task and excluded irrelevant regions from the decision mechanism. The results suggest our sparse BagNet model can be a useful tool for clinicians as it allows efficient inspection of the model predictions and facilitates clinicians' and patients' trust.

**Keywords:** interpretability, sparse activations, diabetic retinopathy

## 1. Introduction

While machine learning (ML) tools have been approaching expert-level performance in many medical imaging tasks thanks to progress in deep learning (De Fauw et al.; Shen et al., 2019; Mahoro and Akhloufi, 2022), they lack interpretability thereby posing ethical concerns (Grote and Berens, 2020) and preventing wide adoption in clinical practice (Teng et al., 2022). Deep ML models are most commonly explained by identifying image regions that influence the output of the trained model with post-hoc saliency maps (Simonyan et al., 2013; Zhou et al., 2016; Springenberg et al., 2014; Selvaraju et al., 2020). However, using saliency maps has been recently shown to be problematic for medical images as they only poorly localize disease-related lesions and are highly variable (Arun et al., 2021; Saporta et al., 2022). Furthermore, they do not provide actionable insights, given that they do not directly reflect the network's actual decision mechanisms. Instead, inherently interpretable models could provide a path forward for safety-critical tasks (Rudin, 2019), but few such models achieve sufficiently high prediction accuracy at the same time. In particular, classical linear models perform poorly when directly applied to medical images.

In this paper, we develop an inherently interpretable deep learning model that combines the feature extraction capabilities of deep neural networks with the advantages in interpretability of sparse linear models. Our model is especially suited for clinically relevant tasks which require identifying and characterising small lesions or other anomalies in large search regions. Examples of such tasks include screening for certain retinal diseases, breast or lung cancer. We focus here on the detection and grading of Diabetic Retinopathy (DR) on retinal fundus images.

DR is a microvascular complication of diabetes characterized by the progressive presence of one or more small retinal lesions such as microaneurysms, hemorrhages, or hard and soft exudates (ICO, 2017). It is the leading cause of blindness in the working-age population and the third leading cause of visual impairment worldwide, and early diagnosis and treatment can slow its progression (ICO, 2017; Wong et al., 2018). It is therefore recommended that diabetes patients undergo regular monitoring, and ML could facilitate mass screening and help clinicians use their time more efficiently (Ting et al., 2016).

Numerous high-performing black-box DR detection methods have been proposed (Rao et al., 2020; Alyoubi et al., 2020; Tavakoli and Kelley, 2021; Huang et al., 2021). For such methods, interpretation is mostly aided by saliency maps (Wang and Yang, 2019; Chetoui and Akhloufi, 2020) or the generation of counterfactual images (González-Gonzalo et al., 2020; Boreiko et al., 2022). A more interpretable approach for detecting DR is a multiple-instance learning model which combines features extracted from different image patches with attention weights (Papadopoulos et al., 2021). These weights can be visualised as a heatmap showing the contribution of different image regions to the prediction. Although all these methods provide some visual evidence of suspicious image regions, the saliency maps do not directly explain the decision mechanism, making their interpretation unintuitive. Further, they are often too cluttered and dense to be useful as feedback for clinicians and may be too coarse to identify small lesions.

We overcome these key limitations and propose a model for DR detection and grading which performs comparably to state-of-the-art models, despite being interpretable-by-

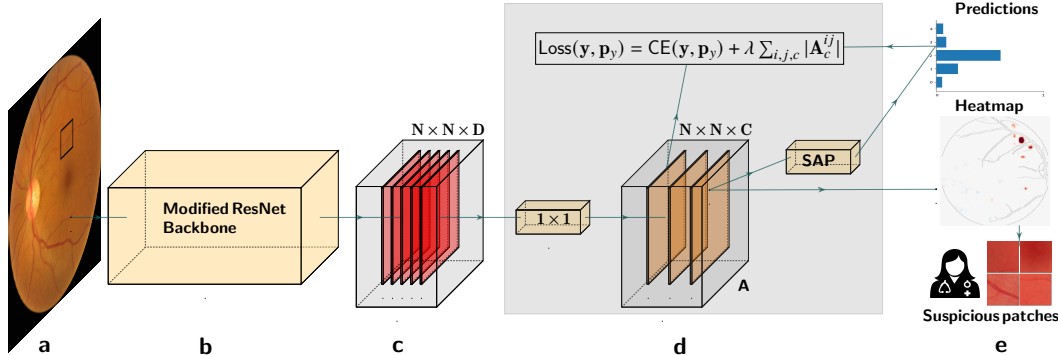

Figure 1: Overview of the proposed interpretable model. **(a)** Input image. The black patch illustrates the small receptive field of **(b)** the modified ResNet-50 backbone. **(c)** The resulting feature map of size $N \times N \times D$, where $D$ is the number of features. **(d)** The class evidence map **A** is obtained with $C$ kernels of size $1 \times 1$ where $C$ is the number of classes. A sparsity constraint can be placed on **A**. **(e)** Top to bottom: class probabilities obtained by spatial average pooling and softmax; example class evidence map; suspicious input patches based on the heatmap.

design[1]. Our approach is based on a bag-of-local-features model (BagNet) (Brendel and Bethge, 2019), which has already been shown to be effective in ophthalmology (Ilanchezian et al., 2021). The BagNet relies solely on local evidence, which makes it relevant for classification or detection tasks where the regions of interest are very small and distributed over the image, as is the case in DR. We propose straightforward but effective changes to the BagNet model which lead to fine-grained class evidence maps which directly and correctly reflect the neural network's decision mechanism. Importantly, our approach allows us to enforce sparse heatmaps, which further aids interpretability and allows the model to precisely identify disease related image regions.

## 2. Developing an interpretable-by-design disease classification network

### 2.1. Backbone architecture and baseline model

We used a BagNet (Brendel and Bethge, 2019) as a baseline classification model. The BagNet is a variant of ResNet-50 (He et al., 2016) and is obtained by replacing many $3 \times 3$ convolutions with $1 \times 1$ convolutions and reducing the strides. This leads to a final feature map **F** of size $N \times N \times D$ where $D$ is the number of feature channels (Fig. 1c, typically $D = 2048$). Spatial average pooling reduces these features to $1 \times D$, and a linear layer then provides the final prediction logits **l** of size $1 \times C$ where $C$ is the number of classes.

These architecture modifications in the BagNet have two effects: First, due to the replaced filters, each image pixel has an effective receptive field of size $q \times q$ in the final feature layer. Therefore, the model makes its predictions based on small image patches of size $q \times q$, implicitly. Secondly, reducing the stride in the convolutional layers prevents

---

1. Our code is available at https://github.com/kdjoumessi/interpretable-sparse-activation

downsampling effects and results in relatively high-resolution (i.e. fine-grained) feature maps compared to the original ResNet.

## 2.2. Enhancing the architecture with an interpretable decision-making stage

To interpret predictions in the BagNet model described above, one cannot directly examine the final feature maps (Fig. 1c), as these represent high-dimensional features rather than class evidence at each location. Rather, it is necessary to construct activation maps with multiple forward passes for individual image patches of size $q \times q$ since the original BagNet architecture is not fully convolutional.

Therefore, we introduced class evidence layers to obtain actual class evidence maps $\mathbf{A}_c$ per class $c$ with a single forward pass and make the local information representation explicit. To this end, we reorganised two network operations. As the spatial average pooling and the final fully connected layer are sums, they can be swapped without affecting the final logits:

$$\mathbf{l}_c = \sum_{d=1}^{D} w_{dc} \left( \sum_{i,j \leq N} \frac{1}{N^2} \mathbf{F}_d^{ij} \right) = \sum_{i,j \leq N} \frac{1}{N^2} \left( \sum_{d=1}^{D} w_{dc} \mathbf{F}_d^{ij} \right) = \sum_{i,j \leq N} \frac{1}{N^2} \mathbf{A}_c^{ij}. \tag{1}$$

Importantly, Eq. 1 can be implemented by replacing the (swapped) FC layer by a $1 \times 1$ convolution with $c$ output channels. The final class-wise evidence maps $\mathbf{A}_c$ directly represent the contribution of individual input patches to the final prediction. The final class score is then obtained by simple spatial averaging (Fig. 1d), resulting in a $c$-dimensional logits vector. Applying the softmax function finally leads to the class probabilities $\mathbf{p}_y$ (Fig. 1e).

## 2.3. Introducing sparsity constraints on class evidence maps

We found that the original BagNet produces dense heatmaps with many positive and negative activations, indicating that clinically irrelevant input patches contribute to the prediction. This behavior makes it difficult for a human to discern how the prediction was formed and to efficiently verify its correctness.

By introducing explicit class evidence layers (Sec. 2.2) we can directly place constraints on the class evidence map containing per-patch scores (Fig. 1d) to induce spatial sparsity. To achieve that, we propose to place an $\ell_1$ regularisation constraint on the class evidence maps $\mathbf{A}_c$, leading to the following loss function:

$$\text{Loss}(\mathbf{y}, \mathbf{p}_y) = \text{CE}(\mathbf{y}, \mathbf{p}_y) + \lambda \sum_{i,j,c} |\mathbf{A}_c^{ij}|. \tag{2}$$

Here, CE denotes the cross-entropy and $\mathbf{y}$ are the reference class labels. The sparsity of the activation maps depends on the hyperparameter $\lambda$. Enforcing sparsity in class evidence in this way is not a post-hoc measure, but rather forces the classification model to focus on the most relevant image regions. This is particularly suitable for tasks such as DR grading where the detection and characterisation of few lesions in the image is sufficient for an accurate diagnostic result and in line with clinical workflows.

Table 1: Classification performance for referable DR detection on the test set.

|  | Accuracy | AUC | Specificity | Sensitivity |
|---|---|---|---|---|
| ResNet-50 | 0.942 | 0.960 | 0.993 | 0.810 |
| Dense BagNet | 0.936 | 0.957 | 0.991 | 0.779 |
| Sparse BagNet | 0.928 | 0.937 | 1.0 | 0.750 |

### 2.4. Advantages of the new architecture and use in a clinical workflow

The proposed modification of the architecture improves the transparency of the model by providing readily interpretable activation maps which show the contribution of each patch to the final prediction without further post-processing. Furthermore, it provides a different class evidence map for each class in a multi-class scenario, directly showing the contribution of each patch to the classification of the input into that class. Importantly, it does so while being less computationally intensive than the original BagNet.

As we will show below, the class activation maps extracted from the sparse BagNet (Fig. 1d) can be upsampled to the input size and overlaid on the input (Fig. 1e) for easy visualisation and interpretation by clinicians. Further, based on activation scores from the class evidence map, suspicious patches (Fig. 1d) can be extracted and presented to the clinician for further investigations (see Sec. 3.4). In contrast to classical saliency maps, one can directly and straightforwardly report how strongly each patch contributes to the network's decision. A clinician can use the global prediction, the class evidence maps, and suspicious patches to either strengthen their trust in the model or reject a decision.

## 3. Results

### 3.1. Dataset

We used retinal fundus images from the Kaggle Diabetic Retinopathy challenge (Kaggle, 2015) with reference DR grades ranging from 0 (no DR) to 4 (proliferative DR). We removed poor-quality images from the dataset using an ensemble of EffcientNet models (Tan and Le, 2019) trained on the ISBI2020[2] challenge dataset. The resulting dataset after the quality filtering contained $45,923$ images with class proportions $(0.73, 0.15, 0.08, 0.03, 0.01)$, which we split into training (75%), validation (10%) and test folds (15%). We preprocessed the images by fitting a circular mask to the field of view and cropping its bounding box. All images were resized to $512 \times 512$ and the image intensities were normalised by the mean and standard deviation of the training set. For additional analyses, an experienced in-house ophthalmologist provided detailed lesion annotations on a selection of 15 test images.

### 3.2. Sparse BagNets yield good accuracy on referable DR detection

We first evaluated our method for the clinically relevant case of (binary) referable DR detection (combining class labels $\{0, 1\}$ vs $\{2, 3, 4\}$). We configured the backbone architecture (Sec. 2.1) such that the receptive field size was $q = 33$ as in Ilanchezian et al. (2021). The

---

2. https://isbi.deepdr.org/challenge2.html

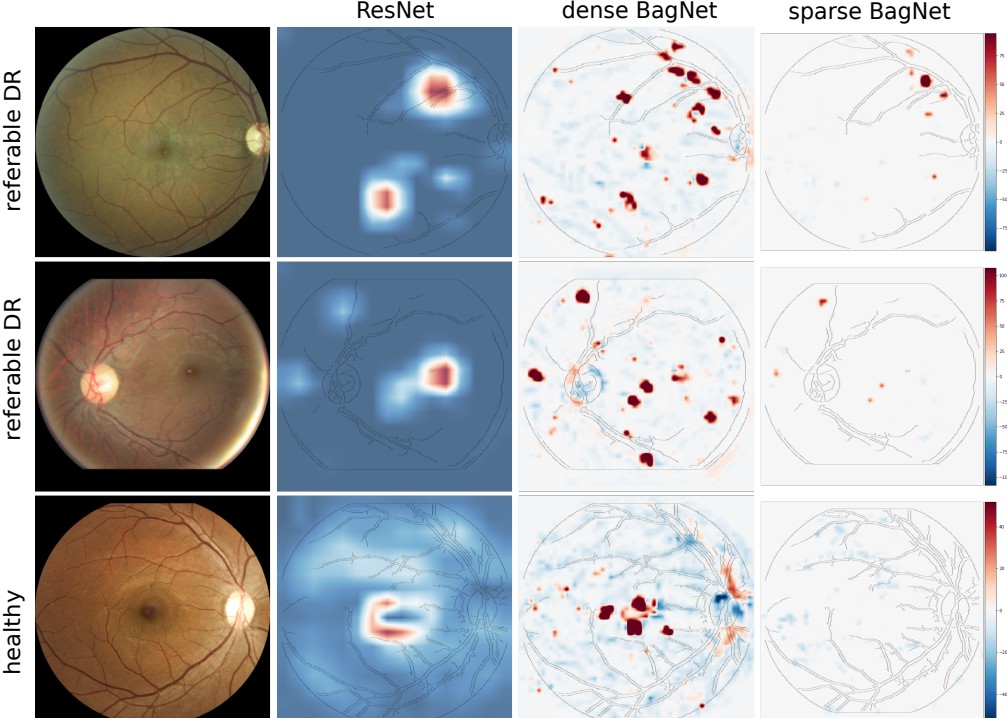

Figure 2: Heatmaps for two example cases with referable DR (top rows) and a healthy case (bottom row). From left to right, heatmaps are shown for the ResNet-50 (using GradCAM), the dense and the sparse BagNet. Red regions provide evidence for the diseased class, while blue regions provide evidence for the healthy class.

regularisation coefficient (Eq. 2) was set to $\lambda = 5 \cdot 10^{-5}$ based on the tradeoff between classification performance on the validation set and sparsity (see App. A). In a realistic application setting, a device manufacturer might define clinically relevant performance thresholds which must be met, and select the maximum sparsity coefficient accordingly.

We compared our sparse BagNet against its dense baseline ($\lambda = 0$) and a ResNet-50 as a black-box state-of-the-art reference. As DR detection has been widely studied classification performance was not our main goal, the training procedures for all models were adopted from Huang et al. (2021), who systematically evaluated hyperparameter choices (see App. B).

We found that the sparse BagNet achieved high accuracy and AUC, which were slightly lower than the respective measures of the dense BagNet and the ResNet50, mainly because of lower sensitivity (Tab. 1). This established that the sparse BagNet was a good candidate for a high-performing interpretable-by-design model, and disease detection performance was not severely hampered by the sparseness penalty on the class activation map.

### 3.3. Sparsity constraints declutter class evidence maps

We next compared local evidence heatmaps indicating important image features obtained from the different models. For ResNet-50, we used the post-hoc technique GradCAM (Sel-

Table 2: Heatmap evaluation on the test set. The first columns show the local-to-global correspondence for images of healthy and diseased eyes, respectively. The third column shows the localisation precision. For all measures, we report mean (std) per image (higher is better).

|  | $\mathbf{r_{LG}^-}$ | $\mathbf{r_{LG}^+}$ | Precision |
|---|---|---|---|
| Dense BagNet | $0.922 \pm 0.03$ | $0.145 \pm 0.06$ | $0.219 \pm 0.1$ |
| Sparse BagNet | $0.991 \pm 0.04$ | $0.374 \pm 0.33$ | $0.791 \pm 0.1$ |

varaju et al., 2020). For the BagNet variants, we used class evidence maps directly from the penultimate layer (Sec. 2.2). The ResNet heatmaps were coarse due to the model's large receptive field and some highlighted regions were similar to regions identified by the dense BagNet (Fig. 2, more examples in App. C). However, as the ResNet's heatmaps do not represent the specific local contribution to the model's decision-making process, we focused on the inherently interpretable BagNet versions for further analysis.

Interestingly, the constraints we imposed on our model led to much sparser heatmaps compared to the original BagNet (see right columns in Fig. 2), showing that the decision was formed from few small regions of the retinal fundus. These regions seemed to be mostly a subset of the salient regions used by the dense model. On healthy images, the sparse model led to an almost complete absence of positive activations, in contrast to the mix of positive and negative evidence suggested by the dense model (see bottom row in Fig. 2).

To assess this quantitatively, we measured how consistently the local class evidence (i.e. the heatmap values) corresponded to the global model prediction. For healthy images, we counted all pixels with negative scores and calculated the ratio of pixels with negative scores among all pixels with non-zero scores, which we call local-to-global correspondence $r_{LG}^-$. This confirmed the qualitative assessment (dense vs. sparse BagNet: 0.922 vs. 0.991; Tab. 2). The same analysis for diseased eyes also further showed a large increase in local-to-global correspondence $r_{LG}^+$ compared to the dense model (0.145 vs. 0.374; Tab. 2). However, on diseased eyes, there remained a large proportion of evidence for healthy tissue, likely because much of the fundus background did not contain any lesions.

### 3.4. High evidence regions correspond to lesions in sparse BagNet

To assess whether the highlighted regions were clinically relevant for diagnosing DR, we quantified the precision of the BagNets' heatmaps at localising DR lesions on the subset of 15 clinically annotated test images. On these, we extracted input patches with positive scores and calculated precision as the proportion of patches that contained a lesion.

The dense BagNet model contained many positive activations in healthy areas without lesions, resulting in low lesion localisation precision (0.219; Tab. 2). In contrast, the sparse BagNet showed considerably increased precision, almost exclusively extracting patches with lesions (0.791). When we visually inspected the patches identified by the sparse model on the annotated images (Fig. 3), we found that almost all patches with positive scores (red boxes, magnified on the right) contained suspicious spots.

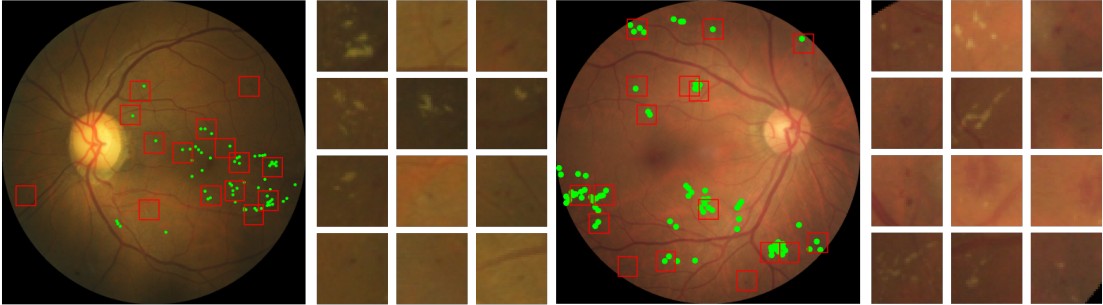

Figure 3: Lesions extracted from the class activation map from the sparse BagNet on two referable DR examples. The green markers indicate reference lesion annotations, whereas the red boxes denote suspicious patches identified by the sparse model (enlarged on the right sorted with decreasing evidence scores).

In fact, when we showed the few seemingly false positive patches (that did not contain an annotation in form of a green marker) to the clinician a second time, she determined that almost all of them likely contain a lesion that had been missed in the original clinical screening.

### 3.5. Sparse BagNets enhance interpretability for multi-class DR grading

Finally, we applied our method to the multi-class setting of DR grading, where individual severity grades were predicted. We used the same training parameters as for the binary task and set the number of output classes to 5. We set the regularisation coefficient of the sparse model to $\lambda = 6 \cdot 10^{-6}$, again choosing an appropriate accuracy trade-off (see App. A). We found that the dense and sparse models achieved comparable accuracy (resp. 0.864, and 0.850) to the baseline ResNet50 model (0.862).

Again, our sparse BagNet model led to more focused heatmaps that were generally consistent with the predicted class (Fig. 4, more examples in App. D). For the example shown in Fig. 4, a clinician retrospectively confirmed that the image appeared healthy except for diffuse bleeding in line with moderate DR in the area highlighted by our sparse model.

Interestingly, further analysis also helped us to uncover a failure mode of the sparse BagNet: We noticed that sparse BagNets always failed to detect grade 3 and most often grade 4 DR cases (Fig. 4). Instead, it tended to classify these cases as moderate DR (grade 2), likely because the sparse BagNet architecture was not designed to detect the larger lesions occurring in these grades.

### 4. Discussion and Conclusion

In this paper, we proposed an inherently interpretable classification model which provides sparse high-resolution class evidence maps. Enforcing sparse activations directly caused fewer input regions to contribute to the classifier decision. We showed that the remaining relevant regions in the sparse model identified lesions with high precision, which is a considerable advance of classical saliency map techniques (Saporta et al., 2022). Further,

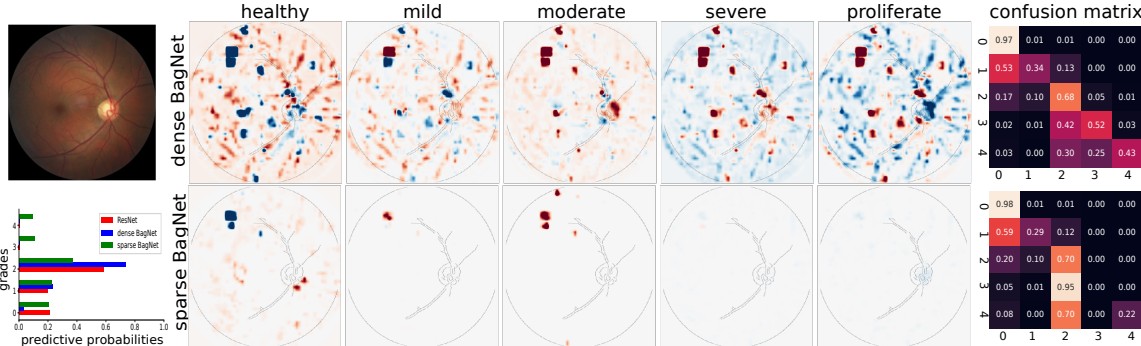

Figure 4: Application to multi-class DR detection shows usefulness of class-specific sparse activation maps of the sparse model over the dense model (bottom row vs. top row; middle column). The example image with moderate DR and predicted probabilities are shown on the left. The confusion matrices (right) show that for the sparse model, severe DR is systematically graded as moderate DR.

healthy images yielded heatmaps with consistently negative evidence. The sparse model was therefore easier and potentially less time-consuming to inspect.

Preliminary feedback from our clinical collaborators suggests that our approach can be a useful tool to verify predictions, understand failure modes of and facilitate their trust in the ML model. Interestingly, they also found the predicted bounding boxes helpful for guiding their attention to subtle anomalies otherwise missed. This suggests future research on ideal ways to let clinicians interact with our model in a human-in-the-loop setting. In a next step, we also plan to apply our approach to other problem settings with local regions of interest such as breast or lung cancer screening.

## Acknowledgments

This work was supported by the German Science Foundation (BE5601/8-1 and the Excellence Cluster 2064 "Machine Learning — New Perspectives for Science", project number 390727645), the Carl Zeiss Foundation in the project "Certification and Foundations of Safe Machine Learning Systems in Healthcare" and the Hertie Foundation. The authors thank the International Max Planck Research School for Intelligent Systems (IMPRS-IS) for supporting Kerol Djoumessi and Indu Ilanchezian. We also thank Sarah Müller for help with image preprocessing, and Murat Seçkin Ayhan for providing an automated quality grading tool.

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

## Appendix A. Effect of the sparsity hyperparameter $\lambda$

The regularisation coefficient $\lambda$ (see Eq. 2) is the hyperparameter that controls the sparsity of the class-specific activation map in the sparse BagNet. It was chosen based on a tradeoff between performance on the validation set according to each task (Fig. 5). Specifically, we manually selected the highest sparsity coefficient for which the performance did not drop too strongly.

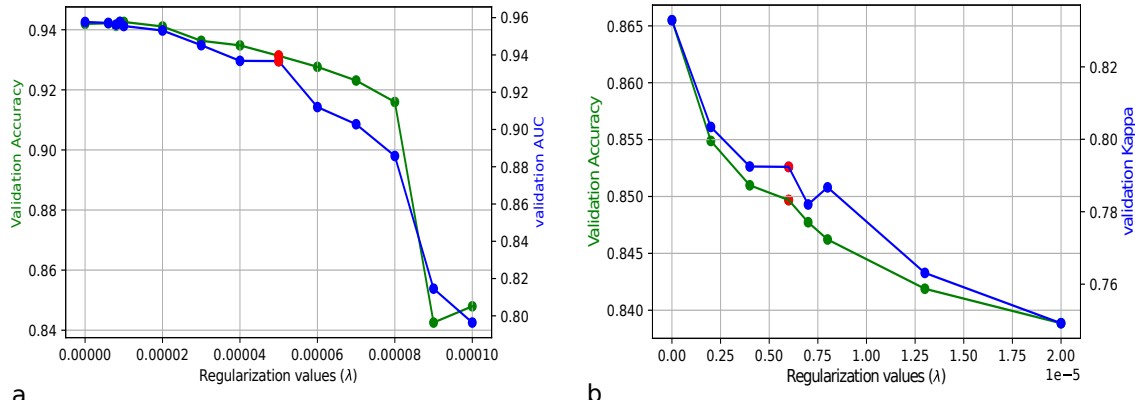

Figure 5: Comparison of validation performance with different regularization values. **(a)** The regularization coefficient $\lambda$ affects the AUC and accuracy on the binary referable task. **(b)** Same as **(a)**, but for the multiclass task with accuracy and kappa. The red points indicate the selected values, which are a trade-of between sparsity and accuracy.

## Appendix B. Training details

We adopted the training regime and hyperparameter choices optimised by Huang et al. (2021), who systematically evaluated relevant hyperparameters for DR detection on fundus images, such as the influence of data preprocessing and augmentation, optimiser and learning rate configurations. The final settings are described below.

We performed data augmentation during training by flipping, rotating, randomly cropping, and translating with a given probability. We also used Krizhevsky color augmentation (Krizhevsky et al., 2017), as suggested by Huang et al. (2021).

To train the networks, we used the cross-entropy loss (unless specified otherwise) as the objective function with the SGD optimizer where the initial learning rate was set to 0.001. Next, the cosine learning schedule was used and the minimum learning rate was set to 0.0001. We also used Nesterov's momentum (Nesterov, 1983) with a constant momentum factor of 0.9 with a weight decay of 0.0005 for regularization.

Models were initialized with weights obtained on the ImageNet and fine-tuned on the Kaggle dataset for 100 epochs with a mini-batch size of 8. The best model was saved on the validation set depending on the task (binary referable DR detection or multiclass DR grading).

For simplicity, and as our goal was not to push the boundaries of classification performance, we did not incorporate features from opposite eyes for image grading (as suggested by Huang et al. (2021)), and also omitted ensembling multiple models.

## Appendix C. Additional examples of class evidence maps

An additional selection of class evidence maps for correctly classified and misclassified examples is provided in Fig. 6.

## Appendix D. Additional results for multiclass setting

The classification performance of our sparse model was comparable to its dense baseline and the ResNet (see Tab. 3). The multiclass task (Fig. 7) shows the advantages of having class-specific activation maps: For example in the last row, small lesions are detected arguing for moderate DR, but larger deteriorations towards the edge of the image argue for proliferate DR. .

|               | Acc.  | Kappa |
| ------------- | ----- | ----- |
| ResNet-50     | 0.862 | 0.826 |
| Dense BagNet  | 0.864 | 0.830 |
| Sparse BagNet | 0.850 | 0.780 |

Table 3: Comparison of the classification performances on multiclass DR detection between the proposed approaches and the baseline ResNet-50 model on the test set.

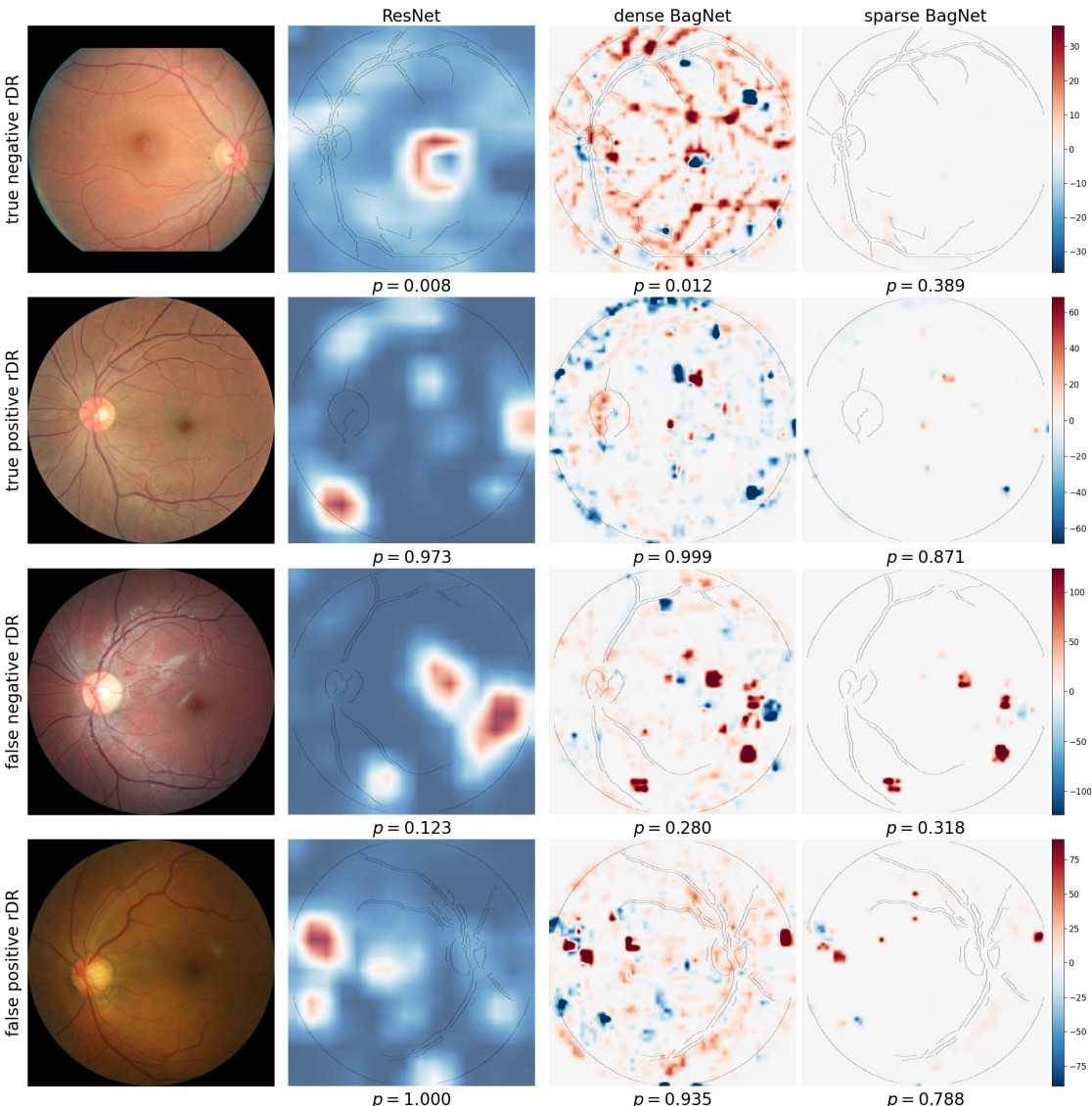

Figure 6: Heatmaps for example of correctly and misclassified cases with referable and healthy DR images. From left to right, heatmaps are shown for ResNet-50 (using GradCAM), the dense and sparse BagNet. Below each heatmap, we also show the predicted probability for referable DR.

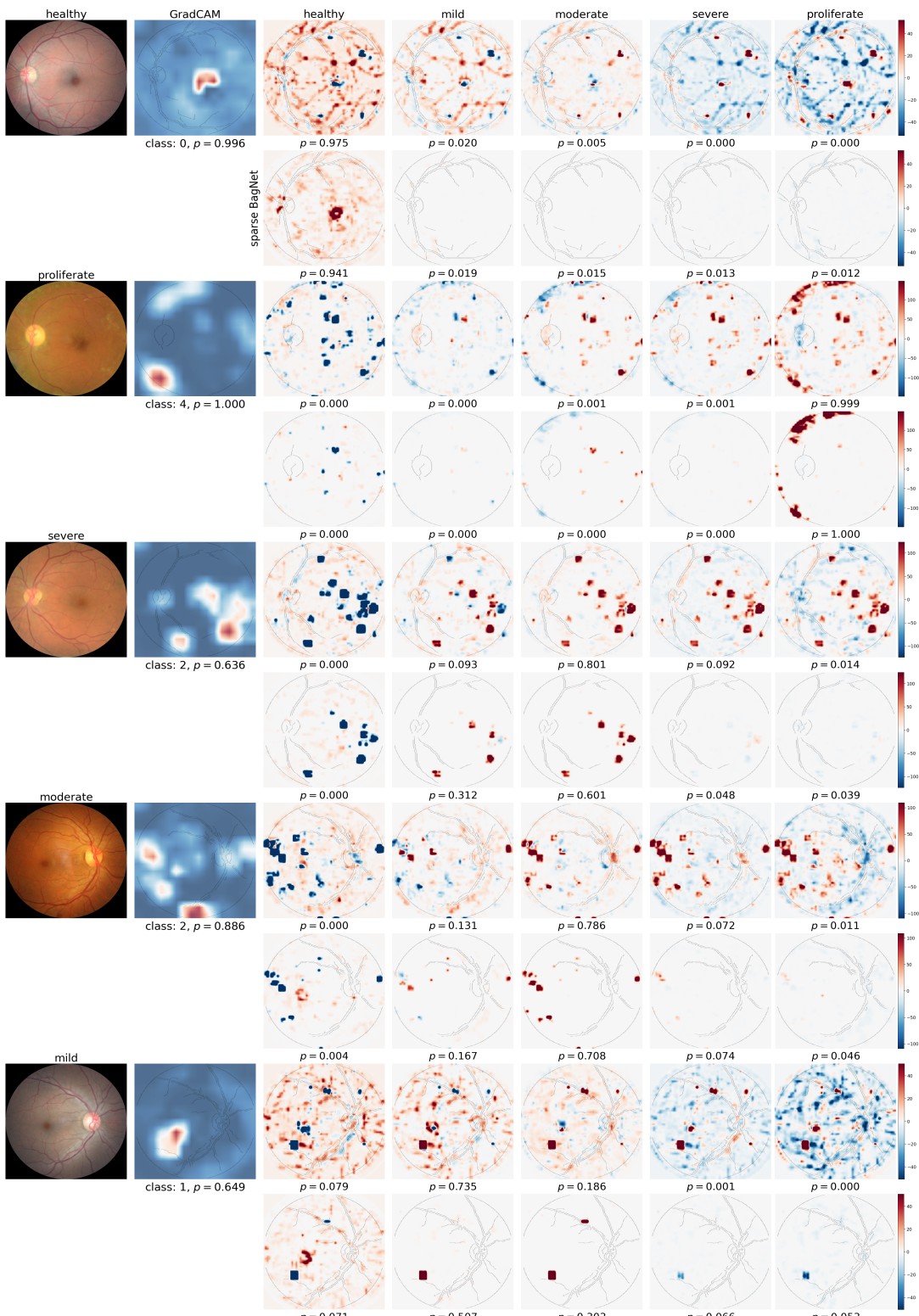

Figure 7: Multi-class evidence maps for images of different DR grades. From left to right, we show the fundus image, the ResNet-50 heatmap (using GradCAM), and the class-specific map for different grades for the dense (top) and sparse (bottom) BagNet. Below each heatmap, we also show the predicted probability for each class, as well as the predicted class and probability for the ResNet-50 model.