# OpenReview forum: "Sparse Activations for Interpretable Disease Grading"
_MIDL.io/2023/Conference — MIDL 2023 Oral_

### Official Review · Reviewer_MkKG · 2023-02-05

**Confidence:** 4
**Preliminary Rating:** 3
**Recommendation:** Poster

**Summary:**

This paper introduces a modification to the BagNet architecture to increase model interpretability. More specifically a sparse decision-making stage is used instead of the final layers in BagNet. The modified model is validated in the detection of Diabetic Retinopathy through the identification of lesions in fundus images. In terms of performance, the proposed model does not outperform other models, but the interpretations provided by the model are sparser.

**Strengths:**

Introducing the sparse decision-making module to the BagNet model architecture is an interesting idea. The final model has the potential for interpretable discovery in the medical imaging domain. Particularly the sparsity constraint could be very relevant in enabling domain-specific interpretations.

**Weaknesses:**

The patch-based method used here is limiting because the model cannot consider more global-level features. This is evident from how it cannot detect patches containing larger lesions. Also, the model does not perform well in multi-class DR detection tasks, possibly because of limited power in characterizing global-level features. Additionally, the paper lacks a thorough discussion of how hyperparameters are selected (except the sparsity constraint). Finally, the quantitative assessments are not reported properly and need confidence measures when appropriate.

**Deanonymize Review:**

no

**Paper Type:**

both

**Questions To Address In The Rebuttal:**

The proposed modified model could be further combined with other modules to address the shortcomings regarding global feature selection. This may increase the performance of multi-class DR detection tasks. The paper also needs a detailed discussion of how hyperparameters are selected.

---

### Official Review · Reviewer_qdsP · 2023-02-05

**Confidence:** 4
**Preliminary Rating:** 5
**Recommendation:** Poster

**Summary:**

The authors present an extension to the bag-of-local-features model (BagNet), which directly estimates sparse evidence maps. By design, this method allows the direct visual interpretation of network predictions, i.e., which patches contributed towards the final classification. The method was evaluated on the task of detecting and grading diabetic retinopathy (DR). The results indicate that the proposed method has comparable performance to a standard ResNet50 model, while being significantly more interpretable.

**Strengths:**

- The paper is very well written and a pleasure to read
- The paper is particularly interesting since interpretability is especially important in any medical setup. The method is especially interesting since the interpretability is straightforward while it does not constrain the model capacity too much
- The adaptations to BagNet are easy to implement and could be easily transferred to other models
- The authors put significant effort into thoroughly evaluating the method
- The authors support all claims with quantiative results and high-quality visual illustrations
- The authors will share the source code

**Weaknesses:**

- It is not perfectly clear why the same evidence maps cannot be achieved by the original BagNet
- It is not perfectly clear how the sparsity hyperparameter was selected
- The authors did not comment on the class occurrences in the dataset
- The authors did not discuss the impact of the aggregation function onto the results
- The visual examples shown in Figures 2, 4, 6, 7 could be further improved

**Deanonymize Review:**

no

**Detailed Comments:**

This paper is very well written and the topic of making neural network predictions interpretable is particularly interesting and highly relevant to the MIDL community. The authors did a great job in thoroughly evaluating and presenting their results. All claims are supported by quantitative and qualitative results. The main content of the paper is very well supported by the supplementary material. There are, however, few questions and concerns:

1. It is not perfectly clear why the same evidence maps cannot be achieved by the original BagNet. The authors mention in Section 2.2: "Rather, it is necessary to construct activation maps with multiple forward passes for individual image patches of size q × q.". However, as shown in Eq. 1, the evidence maps are just the weighted feature maps of the original BagNet. Consequently, a simple multiplication (or 1x1 convolution) of the activation maps is necessary and not multiple forward passes. Could the authors please share some more light into the differences between the original and sparse BagNet? Could the authors also comment on why and how much more computationally performant the proposed method is?

2. What is the distribution of classes in the dataset? In case of severe class-imbalance, could the authors please provide some insights into how this may impact the results?

3. In addition to #2, it seems to me that optimal network performance cannot be achieved by the global average pooling. There are several evidences in the experiments:

    - (1) Binary Classification: As the authors also identified, even diseased images may show a high ratio of healthy / unaffected patches (see ratios in Table 2 & closely look at the evidence maps of Figure 2 - One note: it may be worth showing the evidence maps with the same color scale and/or a color map that shows a different color value for small evidence values and another color for values close to zero). Consequently, naive averaging of the patch-wise evidences will have a bias towards negative values and thus a bias towards healthy prediction. In addition, it looks like the impact of the L1 regularization leads to an overconfident network in healthy patches, which is worsening the situation. It may be worth studying the classification rate of healthy / diseased cases based on the fraction of non-zero evidence pixels being positive / negative and for various thresholds as an alternative to average pooling.

   - (2) While I am not an familiar with DR grading, I would also suspect that it is particularly challenging to infer the DR grade from a single patch - I would even assume that severe cases show a multitude of lesions with different characteristics. Under this assumption and particularly assuming that severe cases show fewer patches with severe lesion, the simple averaging per class channel will lead to lower average values for the severe cases comapred to mild disease progression. This may be particularly worsened if the class distribution is imbalanced and grad 3 is underrepresented.

   - It would be extremly interesting to hear the authors' opinion on this hypothesis.

4. Figure 6 & 7: It is not clear whether all networks correctly classify / misclassify the images. Could the authors please add the probability score for each image & network? Please also adapt the color bars to show the range of the evidence map

5. The authors mention that this way of interpreting network results could help in understanding failure modes and facilitating the trust into such models. Could the authors please share their thoughts on how this information could then be again used to improve such models?

Minor comments:
- In Section 2.4: a closing bracket is missing -> (see Sec. 3.4)
- Figure 3: This is an extremely powerful visualization and I am very happy to see such a comparison. Since the authors show extracted patches from the evidence map, I believe one possible extension of this analysis is applying GradCAM on the evidence map to obtain a heat map for the presented patches. Apart from possibly additional information for clinicans, it would be interesting to see whether the heat map maxima are near the ground truth annotation.

- Figure 4: Comparing the activation maps to the bar plot showing the predictive probabilities, it is not evident why the dense BagNet and sparse BagNet infer identical probabilities. Could the authors please double check that there is no error? If possible, could the authors also show the annotation by the clinician for this case?
- Page 6, Section 3.2: Please clarify that its the class labels that are combined.
- Choice of sparsity parameter: The authors mention that the hyperparameter was chosen based on a trade-off. Could the authors please share some insights what specific criterion was used?
- The last page is empty and could be removed

**Paper Type:**

methodological development

**Questions To Address In The Rebuttal:**

I would kindly ask the authors to address the questions and concerns mentioned in the preceeding sections. Since the paper is already in very good shape, I would put most importance onto #2 and #3 of the detailed comments section.

---

### Official Review · Reviewer_2Doa · 2023-02-05

**Confidence:** 4
**Preliminary Rating:** 4
**Recommendation:** Poster

**Summary:**

This paper describes experiments with the Kaggle-DR set, which is a large dataset with 2D color fundus images that are graded for diabetic retinopathy from 0 to 4. Many papers have used this data set. The authors want to develop a method that is ‘inherently interpretable’. The propose an architecture which classifies (while the task they consider here is grading, not classifying) and identifies a set of sparse (small, and a limited number) regions in the image that are important for classification.

**Strengths:**

Seems solid work.
The authors propose a method that can give more sparse heatmaps, for tasks where the overall class of an image is determined by a few small regions in the image. In certain situations, such sparse heatmaps may be more useful than heatmaps that give some level of response almost everywhere or provide large blobs only, as is typical for the earliest generations of heatmaps.

**Weaknesses:**

Code is not shared, but only announced to be shared.
The paper lacks convincing experimental evidence that the proposed sparse maps are superior to other methods
The task in this dataset is a grading task, but the method treats it as a classification task. This is problematic (see below)
It is not exactly clear which images were used for the experiments
The proposed method is not as accurate as other methods, this is may be problematic (see below)

**Deanonymize Review:**

yes

**Detailed Comments:**

There are already many methods for producing saliency maps. I’m not sure if this method adds much; no convincing experiments that show that this way of extracting relevant regions outperform other methods. I could not follow the description of experiments that resulted in Table 2. In the end it seems the results are best summarized on page 8, clinicians gave feedback that the proposed approach can be a useful tool.

The task addressed is a grading task, and grades are determined by the number of lesions. It is unclear if independently highlighting regions makes sense in this scenario. If an image is positive when it contains more than 5 trees and you process an image with 17 trees, should all 17 be highlighted? Or any random subset of 5? The work of Gonzalez, https://doi.org/10.1109/TMI.2020.2994463 addresses this issue explicitly, identifying regions in an iterative manner, using the same dataset.

The authors selected a subset of Kaggle-DR, making it hard to compare to other methods, but show that their sparse BagNet performs worse than a standard ResNet-50 (table 1). It is challenging to claim one finds better interpretable features if the performance of the underlying method is less good.

Code is not shared, but announced to be shared. Research from Umeå University showed (https://arxiv.org/abs/2210.11146) that (MIDL) papers regularly promise to share code but ultimately the code is not shared or the code shared is not of high-quality. In my opinion, code should be shared with reviewers during the review period and conferences should require that code is really shared.

Assessing methods that produce heatmaps to interpret DL classification results is difficult. It would be great if the authors could sketch a way to properly compare the quality of such heatmap methods.

Typo effcientnets


**Paper Type:**

methodological development

**Questions To Address In The Rebuttal:**

The above lists several questions. All questions should be addressed. Sharing the code is the most important to me. Ideally all code is shared, for the new method and for all experiments that are presented in the paper.

---

### Meta-Review · Area_Chair_Mtzg · 2023-02-22

**Recommendation:** Accept (Poster)
**Confidence:** 5

**Metareview:**

Sparse Activations for Interpretable Disease Grading

The consensus is on acceptance, ranging from borderline positive to strong accept. The contribution is towards producing sparse activation maps instead of full-image maps. Clarification is needed on why the proposed sparse method improves over, for instance, the original bag of local features approach. Due to the noted positive appreciation and given the global ranking of submissions, the recommendation is leaning towards acceptance.

Recommendation towards Acceptance.